## [Transparent Peer Review File · Communications Biology]

Rocking-induced sleep enhancement promotes motor learning through transcriptional and synaptic remodelling

Corresponding Author: Dr Michele Bellesi

Version 0:

Reviewer comments:

Reviewer #1

(Remarks to the Author)

The research question addressed here is both important and timely. Although some evidence for sleep extension has been shown with a rocking motion during sleep, whether or not vestibular stimulation induced by rocking impacts motor memory is not known. Therefore the investigation by Simayi et al. is both worthwhile and pertinent to the sleep/memory field at large. Addressing the following points will serve to improve the manuscript:

- In the Figure 1 experiment, was “Rocking ON” for the entire 24 hr recording period across both Lights On and Lights Off? If so, please clarify in the legend or Fig1A schematic.
- Figure 1C and 1D: should Y-axis be % of 12 hour lights on and lights off period respectively?
- Figure 3B: Is there a minimum duration used to define what the maximum running speed was per session or was absolute maximum running speed used as the metric irrespective of how long that running speed was maintained.
- Related to both Figure 3B/3D? average bout length of running is a nice control comparison. Is there another measure that captures learning such as average maintained speed during each day to compare between sleep conditions? Something like cycling “pace” that ignores ramp up/breaks but averages across maintained running.
- Figure 3E: please put the correlation stats on the figure so we don't have to hunt for it in the Results text.
- Figure 4B: for audience interpretation, can the authors illustrate when the rocking is On during the activity plot X-axis? Presumably for 4 hours at the start of the gray bar.
- Figure 5 design: to facilitate interpretation of the in transcriptional changes, when was tissue collected for RNA seq in relation to the rocking protocol/# of days or # of motor testing sessions? Was tissue collected immediately harvested after only the rocking or after both rocking and motor?
- For the GO analyses, how should the audience interpret what the X-axis “Signal” means? Is that a proportion of signal relative to sleep control, ie: SE/S?
- Figure 7 and mention of this manuscript with a nice control analysis of assessing the effect of rocking only on synapse type/number, but did the authors look at other brain regions that may change (ie NOT motor cortex)? Synaptic downscaling during sleep would lead one to hypothesize less glutamatergic synapses post-SE, at least in the hippocampus.

Discussion

- I would also like to see more discussion of the control experiment whereby increased synaptic measures were not observed following rocking alone, but “required” motor learning following sleep enhancement. Might brain region location play a role?

Reviewer #2

(Remarks to the Author)

Reviewer #3

(Remarks to the Author)

Major Claims

The manuscript investigates how vestibular stimulation (rocking) influences sleep and subsequent behavioral outcomes,

with additional analyses on synaptic density markers. The central claims are that rocking modulates sleep-wake patterns and improves memory-related measures. While the idea of rocking as a modulator of sleep is intriguing and of potential translational interest, aspects of the claims lack novelty without strong mechanistic grounding. Several prior studies should be cited to contextualize rocking and vestibular contributions to sleep.

Convincing Evidence

The data provide preliminary evidence but do not convincingly support the full conclusions.

- The pilot PSG dataset is important but limited in scope; most conclusions are drawn from behavioral inferences that could be confounded.
- Tapping during certain conditions acts as an uncontrolled arousal cue, complicating interpretation of rocking-specific effects.
- Additional controls (e.g., sham vibration/noise, habituation analysis across days) and colocalization markers (VGLUT1 + PSD-95) would strengthen claims about synaptic plasticity.

Impact on the Field

If clarified and strengthened, this work could spark interest in non-pharmacological interventions to improve sleep and learning. However, as currently presented, the conclusions are not robust enough to significantly alter prevailing thinking in the field. The potential translational impact is promising but requires stronger mechanistic and methodological rigor.

Statistical Analysis

Statistical approaches do not appear fully aligned with the design. Given repeated measures across the day and possible cage-level clustering, simple t-tests and ANOVAs are not sufficient. Mixed-effects models would be more appropriate. Effect sizes, confidence intervals, and exact p-values should be consistently reported. Without this, the strength of the evidence is difficult to assess.

Reproducibility

The methodological description is detailed in some areas (e.g., rocking apparatus) but insufficient in others:

- Rocking parameters (1 Hz, ± 20 mm displacement) are not justified or compared against alternatives.
- Housing conditions (group vs. individual) and cage placement on the platform could strongly affect results but are not thoroughly described.

Other Concerns

- Habituation is not considered; 11 days of exposure may lead to reduced efficacy after 7 days.
- Recovery sleep/homeostatic rebound may explain some effects rather than rocking itself.

Suggestions for Clarity and Organization

- Clarify the rationale for parameter choices (frequency, duration, displacement).
- Reorganize Results to emphasize consistent findings, grouping nonsignificant results succinctly.
- Expand the Discussion to explicitly address alternative explanations (arousal cues, circadian phase, homeostasis).

Transparency

I prefer to remain anonymous for this review.

Reviewer #4

(Remarks to the Author)

I co-reviewed this manuscript with one of the reviewers who provided the listed reports. This is part of the Communications Biology initiative to facilitate training in peer review and to provide appropriate recognition for Early Career Researchers who co-review manuscripts.

Version 1:

Reviewer comments:

Reviewer #1

(Remarks to the Author)

The authors have done a fantastic job in responding to our commentary, and I have no further concerns.

Reviewer #2

(Remarks to the Author)

I appreciate the authors' responses and am satisfied with their revisions.

Reviewer #3

(Remarks to the Author)

The authors have responded thoughtfully and extensively to my previous concerns, substantially strengthening the manuscript. Most major issues have been addressed. However, some responses rely on justification rather than additional evidence, and a few conceptual and methodological limitations remain that should be more clearly acknowledged.

I appreciate the authors' careful and comprehensive responses to my previous comments. The manuscript has been substantially strengthened, particularly through improved statistical reporting, clearer hypothesis testing, and more appropriately the synaptic and behavioral findings.

while the control experiments substantially limit nonspecific arousal or activity confounds, the absence of a dedicated sham-rocking condition modestly constrains inferences about vestibular specificity and should be acknowledged as such.

Overall, the authors have satisfactorily addressed the major concerns raised in the initial review. With minor clarifications regarding these remaining points, the manuscript represents a well-controlled and timely contribution to the literature on sensory sleep enhancement and experience-dependent plasticity.

Reviewer #4

(Remarks to the Author)

I co-reviewed this manuscript with one of the reviewers who provided the listed reports. This is part of the Communications Biology initiative to facilitate training in peer review and to provide appropriate recognition for Early Career Researchers who co-review manuscripts.

REPLY TO REVIEWERS

We would like to sincerely thank all the reviewers for the time and effort they dedicated to evaluating our work. Their constructive and insightful comments have been extremely valuable and have substantially improved the clarity, rigor, and overall quality of the manuscript.

Reviewers' comments:

Reviewer #1 (Remarks to the Author):

The research question addressed here is both important and timely. Although some evidence for sleep extension has been shown with a rocking motion during sleep, whether or not vestibular stimulation induced by rocking impacts motor memory is not known. Therefore the investigation by Simayi et al. is both worthwhile and pertinent to the sleep/memory field at large.

We thank the reviewer for their positive assessment and constructive comments.

Addressing the following points will serve to improve the manuscript:

1. In the Figure 1 experiment, was “Rocking ON” for the entire 24 hr recording period across both Lights On and Lights Off? If so, please clarify in the legend or Fig1A schematic.

Thanks for this suggestion, we added a label in the dark period bar to improve clarity.

2. Figure 1C and 1D: should Y-axis be % of 12 hour lights on and lights off period respectively?

Corrected

3. Figure 3B: Is there a minimum duration used to define what the maximum running speed was per session or was absolute maximum running speed used as the metric irrespective of how long that running speed was maintained.

We thank the reviewer for this useful question. The maximum running speed was not based on a single instantaneous peak but was defined as the highest average speed achieved during any running bout within a given day. Each running bout was required to last at least 5 seconds, ensuring that the maximum speed reflected a sustained running effort rather than a transient fluctuation. We added these details to the methods, which now reads: *For each animal, all running bouts were identified and characterized. A running bout was defined as a continuous period of activity lasting at least 5 seconds, with significant breaks (> 5 seconds) used to separate consecutive bouts. For each bout, the average speed was calculated as the total distance run divided by the bout duration, providing a measure of the animal's running “pace.” The daily average speed was defined as the mean of all bout average speeds, while the maximum speed was defined as the highest average speed achieved in any running bout within a given day, ensuring that the speed was maintained for at least 5 seconds. The total daily distance was computed as the sum of the distances run across all bouts.*

4. Related to both Figure 3B/3D? average bout length of running is a nice control comparison. Is there another measure that captures learning such as average maintained speed during each day to compare between sleep conditions? Something like cycling “pace” that ignores ramp up/breaks but averages across maintained running.

We thank the reviewer for this thoughtful suggestion. In addition to maximum speed, we also calculated the average maintained speed, defined as the grand average of the average speeds across all running bouts within a given day. Each running bout was required to last at least 5 seconds, and

significant pauses (>5 seconds) were used to separate bouts, ensuring that the metric reflects sustained running (“pace”) rather than brief accelerations or transitions. The results for average maintained speed closely mirrored those obtained for maximum speed, showing the same group differences across sleep conditions. We have now included both measures in the revised analysis and figure 3, and, as suggested by Reviewer 2, we also added analyses of the number of running bouts and total distance as additional control metrics.

Results

“Motor performance was analyzed by measuring the maximum and average running speed across the 11-day experiment period in both the SE and S groups. Both groups showed a gradual increase in speed, indicating progressive motor adaptation to the complex wheel task. However, the SE group consistently outperformed the S group, demonstrating a faster rate of improvement and reaching peak performance earlier (Figure 3B-C). To quantify learning, we calculated the % improvement, defined as the ratio between maximum (or average) speed on the last and first day of wheel use. SE mice showed significantly greater improvement from baseline compared to S mice (unpaired t-test: maximum speed, $P = 0.016$; average speed, $P = 0.04$; Figure 3D-E).

We next examined the relationship between sleep and motor learning by correlating total sleep duration with % improvement. Positive correlations were observed in both groups for average and maximum speed (max: SE: $r = 0.52$, $P = 0.048$; S: $r = 0.48$, $P = 0.12$; avg: SE: $r = 0.62$, $P = 0.03$; S: $r = 0.45$, $P = 0.097$; Figure 3F-G). In contrast, the fragmentation index, used as a measure of sleep consolidation, showed a negative but non-significant correlation in both groups (SE: $r = -0.2$, $P = 0.47$; S: $r = -0.24$, $P = 0.46$; Figure 3H). To formally assess whether these relationships differed between groups, we fitted a linear model including a Group \times Sleep interaction. The interaction was not significant (max: $\beta = 30.68$, $SE = 152.15$, $t = 0.20$, $P = 0.84$; avg: $\beta = -30.5$, $SE = 131.58$, $t = -0.23$, $P = 0.82$), indicating similar slopes between SE and S groups for both maximum and average speed. Together, these results suggest that longer sleep duration was similarly associated with greater motor learning in both conditions.

Finally, to assess whether the enhanced learning in SE mice was driven by increased wheel use, we analyzed multiple measures of running activity, including the average bout duration, the average number of bouts, their product (bout duration \times number), and the total distance traveled. No significant difference was observed between groups in average bout duration (unpaired t-test: $P = 0.36$, Figure 3I), but SE mice exhibited a higher average number of bouts than S mice (unpaired t-test: $P = 0.004$, Figure 3J). However, neither the bout duration \times number nor the total distance differed significantly between groups (bout duration \times number: unpaired t-test: $P = 0.13$; distance: unpaired t-test: $P = 0.18$, Figure 3K-L). Importantly, neither of these measures correlated with learning performance (bout duration \times number: SE: $r = -0.13$, $P = 0.64$; S: $r = -0.04$, $P = 0.91$; distance: SE: $r = -0.08$, $P = 0.78$; S: $r = 0.15$, $P = 0.64$; Figure 3M-N), whereas sleep amount did.

These findings indicated that rocking during sleep facilitated motor learning on the complex wheel task, and this effect was primarily related to increased sleep, rather than greater time or activity on the wheel.”

Figure 3. Rocking improves motor learning on the complex wheel task.

A. Image of the complex wheel and schematic representation of its irregular rung pattern. **B–C.** Maximum (**B**) and average (**C**) running speed of S and SE mice across the 11-day experiment. **D–E.** Motor learning quantified using maximum (**D**) or average (**E**) speed. **F–G.** Correlation between learning (maximum (**F**) and average (**G**) speed) and sleep amount (normalized to baseline) in S and SE mice. **H.** Correlation between learning (calculated using maximum speed) and sleep fragmentation. **I.** Average bout duration. **J.** Average number of running bouts. **K.** Product of average bout duration \times number of bouts. **L.** Total distance travelled. **M–N.** Correlation between average bout duration \times number (**M**) or total distance travelled (**N**) and learning. For all bar graphs, data represent mean \pm std. S (n = 12), SE (n = 15). * $P < 0.05$, ** $P < 0.01$.

Discussion

“...An important consideration is whether the enhanced performance of SE mice could reflect differences in wheel engagement rather than the effects of sleep itself. Although SE mice initiated running bouts more frequently, the total time spent running and distance covered did not differ significantly between groups. Moreover, these measures did not correlate with learning outcomes, in contrast to the clear relationship observed between sleep amount and performance improvement. Thus, it is unlikely that the improved learning observed in SE mice was simply due to increased physical activity or greater task exposure; rather, the data support the idea that sleep enhancement directly facilitated the acquisition of this complex motor skill..”

5. Figure 3E: please put the correlation stats on the figure so we don't have to hunt for it in the Results text.

Corrected. See Figure 3 in point 4 above.

6. Figure 4B: for audience interpretation, can the authors illustrate when the rocking is On during the activity plot X-axis? Presumably for 4 hours at the start of the gray bar.

Done.

7. Figure 5 design: to facilitate interpretation of the in transcriptional changes, when was tissue collected for RNA seq in relation to the rocking protocol/# of days or # of motor testing sessions? Was tissue collected immediately harvested after only the rocking or after both rocking and motor?

We thank the reviewer for this comment. Tissue collection was collected between 8 and 9 am on the 12th day, after the end of the last session of complex wheel for both S and SE mice. This information is now clearly reported in the methods.

“S and SE mice were sacrificed between 8:00 and 9:00 AM on the 12th day, following completion of the final complex wheel session...”

8. For the GO analyses, how should the audience interpret what the X-axis “Signal” means? Is that a proportion of signal relative to sleep control, ie: SE/S?

In our GO analysis, the X-axis term 'Signal' represents the enrichment score for each GO term, reflecting how strongly a term is overrepresented in our gene set compared to a background set. It does not indicate a proportion (e.g., SE/S), but rather the relative strength of association between the GO term and the genes of interest. We have clarified this in the figure legend.

“For panels B–D, the x-axis represents the enrichment score (signal), which indicates how strongly each GO term is overrepresented in our gene set relative to the background set.”

9. Figure 7 and mention of this manuscript with a nice control analysis of assessing the effect of rocking only on synapse type/number, but did the authors look at other brain regions that may change (ie NOT motor cortex)? Synaptic downscaling during sleep would lead one to hypothesize less glutamatergic synapses post-SE, at least in the hippocampus.

Thank you for raising this important point. We measured VGLUT1 and VGAT puncta density in the CA1 region (stratum radiatum) of the hippocampus under the condition without wheel access. Quantitative analysis did not reveal any significant differences between the two groups for either VGLUT1 or VGAT. Unlike previous work reporting a reduction in synapse number following sleep compared to sleep deprivation (Spano et al., J. Neurosci., 2019, 39:6613–6625), we did not observe any change between S and SE mice. This discrepancy may have several explanations. In the previous study, the difference was evident only when non-perforated synapses were analyzed, whereas our approach did not distinguish between synapse subtypes. Moreover, our comparison involved two

sleeping conditions, where the expected effect size of synaptic downscaling is likely smaller and may fall below the detection threshold of our method. It is also possible that such differences become apparent only when sleep is directly compared with sleep deprivation. We now include these new results and have expanded the Discussion accordingly (see point 10).

Results

“To determine whether SE induced changes in synapse number outside the primary motor cortex, we measured VGLUT1 and VGAT puncta density in the stratum radiatum of the hippocampal CA1 region, which is critically involved in learning. This region was also selected due to prior evidence of sleep-dependent changes in synaptic density²³. In our analysis, SE did not significantly alter the number of VGLUT1- or VGAT-positive puncta (Linear mixed model: VGLUT1: $P = 0.14$; VGAT (neuropilar): $P = 0.46$; VGAT (perisomatic): $P = 0.84$) in this hippocampal region (Figure J-M).”

Discussion

(see the point below)

10. I would also like to see more discussion of the control experiment whereby increased synaptic measures were not observed following rocking alone, but “required” motor learning following sleep enhancement. Might brain region location play a role?

We have expanded the Discussion accordingly, incorporating also the new hippocampal findings as follows:

Discussion

“...It is also possible that other brain regions are more sensitive to the effects of sleep enhancement when synapse number is considered. For example, a recent study using three-dimensional electron microscopy showed that, unlike in the motor cortex, sleep reduces the number of synapses in the CA1 stratum radiatum relative to short sleep deprivation²³. In contrast, our data from the same hippocampal region did not reveal any difference in synapse density between S and SE mice, suggesting that rocking-induced sleep enhancement does not influence hippocampal synapse number under our experimental conditions. This apparent discrepancy with previous findings may arise from several factors. The earlier study reported effects of sleep only for non-perforated synapses, whereas our confocal approach did not distinguish between synapse subtypes. Moreover, our comparison involved two sleeping conditions, where the potential effect size of sleep-dependent downscaling is likely smaller and possibly below the detection threshold of our method. Finally, it is conceivable that differences in synapse number emerge primarily when sleep is compared with sleep deprivation, rather than between two conditions of normal or enhanced sleep.”...

Reviewer #2 (Remarks to the Author):

Review Comments on Commsbio-25-7685

Rocking-induced sleep enhancement promotes motor learning through transcriptional and synaptic remodeling

Comments for Author:

Simayi and colleagues present a nice paper with a well-thought-out experiment. They identified an important topic of noninvasive sleep enhancement, noting that vestibular stimulation through rocking has been demonstrated to improve sleep. They reason that, if rocking improves sleep, it should also enhance sleep-dependent behaviors and respective neurophysiology, and they set out to measure both these potential outcomes of sleep-enhancement through rocking in mice. The experiment is well-designed, and the analyses

(including the control experiments) are appropriate to the hypotheses proposed, adding to the field's knowledge of sleep-enhancement techniques and sleep-dependent motor learning physiology. The authors overstate their findings a couple of times, but this can be addressed. A few comments, questions, and suggestions below will help improve the paper.

We thank the reviewer for the positive and constructive comments.

1. Abstract – “This improvement in sleep...”. This should be “These improvements” (plural) or it should directly state which improvement. Figure 3 shows that would be normalized duration, not consolidation.

We have corrected these sentences. Thanks.

2. The two sleep outcomes that were improved by rocking were sleep duration and sleep-wake transitions, which the authors label as consolidation. Was the sleep duration increase an increase in the overall sleep period (e.g. a sleep extension), and/or was it due to the reduced sleep-wake transitions? In other words, was there just an improvement in sleep efficiency caused solely by the reduced number of wakeups, or was there both reduced fragmentation and extension of total sleep time?

We appreciate the reviewer raising this point. As we reported in a recent study (Zhang L, et al., *iScience*. 2025 Feb 15;28(3):112036) and now confirm here with EEG, rocking primarily extends the duration of individual sleep bouts and, to a lesser extent, reduces their number. The motion-detection method used for sleep/wake tracking is less sensitive than EEG, as it cannot reliably detect brief awakenings during immobility. As a result, two or more sleep episodes separated by short immobile awakenings may be scored as one continuous period. Nevertheless, motion detection accurately captures transitions between immobility and movement, which we used as a proxy for sleep fragmentation. In our data, rocking increased total sleep duration (time spent immobile) and reduced immobility-movement transitions, consistent with both extended sleep and reduced fragmentation.

3. Related to the correlation between sleep duration increase and learning (Fig 3E), why was this correlation only done with duration when rocking improved both duration and consolidation? That should be justified or the correlation repeated for all sleep outcomes.

The reviewer is correct. In addition to sleep duration, we have now analyzed the correlation with the fragmentation index as a measure of sleep consolidation. This analysis revealed a negative, though non-significant, correlation in both groups of mice. These results are now reported in the revised manuscript. See point 4 of Reviewer 1.

4. Figure 1 and Figure 2 both suggest that the sleep enhancement effect of rocking is apparent after only 24 hours. Why was 11 days of rocking chosen for the motor learning experiments? This should be justified.

We thank the reviewer for this important question. We chose 11 days of rocking for the motor learning experiments because this task requires multiple days of training to reach a performance plateau (as reported in McKenzie IA, et al. *Science*. 2014 Oct 17;346(6207):318-22.). In contrast, EEG recordings were limited to a single day of rocking, as our animal protocol did not allow tethering for more than 48 hours, and we did not have access to wireless EEG systems. The purpose of including EEG experiments was to validate the reliability of our motion detection approach in quantifying accurately sleep and wake modifications following rocking validating the motion-detection approach, rather than to assess the duration of the rocking effect. The decision to apply rocking for 11 days was made to ensure sufficient time for the mice to learn the complex wheel task. As specified in the methods, *mice typically learn to navigate the complex wheel and reach peak running speed within approximately one week*. Extending the training period by a few additional days ensured that all animals had adequate opportunity to learn this complex motor task.

We added a sentence in the methods “*Under normal conditions, mice typically learn to navigate the complex wheel and reach peak running speed within approximately one week. In this study, the training period was extended to 11 days to ensure that all animals, including slower learners, had sufficient time to achieve stable performance and fully consolidate the acquired motor skill.*”

5. “Despite the demonstrated efficacy of both acoustic and vestibular modalities in modulating sleep architecture, the underlying neurobiological mechanisms remain poorly understood.” From the Introduction, this sentence should be removed or clarified. This paper does not examine mechanisms of vestibular sleep-enhancement, but specifically mechanisms of motor learning that are hypothesized to be caused by rocking-enhanced sleep. The very next sentence pivots back to correctly talking about behavioral outcomes of sleep, rather than mechanisms of sleep enhancement, which is appropriate.

We have removed that sentence.

6. Citations 10, and 17 in the following sentence: “Most studies have focused on behavioural outcomes, particularly memory and learning, with mixed and sometimes inconsistent results 10,17” These two citations are only about acoustic stimulation, and most concerningly, citation 10 Stanyer et al, has been retracted. The authors should remove that citation, and should add references of inconsistent behavioral effects of vestibular stimulation, if those exist. If no studies to date have done this, that should be stated.

We thank the reviewer for pointing this out. At the time of writing, the Stanyer et al. study had not yet been retracted; we have now removed it from the references. In addition, we revised the sentence to specifically address vestibular stimulation and clarified that, while only a few studies have investigated its behavioral effects, evidence remains limited.

“Most studies on acoustic stimulation during sleep have focused on behavioural outcomes, particularly memory and learning, with mixed and sometimes inconsistent results ^{16–18}. Evidence that vestibular stimulation may also support memory consolidation is limited to a few studies in humans and flies ^{14,19}”

7. In Results, paragraph 1: The “pilot polysomnographic experiment” as described in the methods, is not described as such in the Results first paragraph. It was unclear that these were the same thing. I suggest the authors align the language between Results and Methods, and especially in Figure 1, which doesn’t label the figure as the pilot/verification experiment. These edits will help with understanding the purpose of the pilot/verification study in figure 1.

Thanks for this suggestion. We have corrected the text to align the language between Results, Methods, and Figure legend.

8. Relatedly, Figure 1 should specify that this is from a separate group of animals that had PSG and rocking for only 1 night, and especially that Wake/NREM/REM staging was performed using PSG data rather than video.

Corrected

9. In Results, section “Rocking during sleep promotes motor learning”, the authors want to determine whether the improved learning rate was due to increased wheel-use or caused by the sleep-enhancement directly. To do so, they measured average wheel bout length and found no significant group differences. This doesn’t answer the question about whether there was a higher total number of bouts, just the average duration of the bouts. So, it remains possible that the SE group was more active (overall more running bouts but of similar duration) during the Dark phase than the S group, which could still account for improved learning.

Thanks for raising this very important point. The reviewer is right that assessing bout number is essential. Following this advice, we calculated the average number of bouts and found it to be significantly higher in SE compared to S. However, average bout number x average bout duration and total distance traveled (both good proxies for overall wheel use) did not differ significantly, although SE mice tended to spend more time on the wheels on average. These findings suggest that SE mice may have had more running opportunities, but this alone cannot account for their superior performance. Critically, neither bout number, nor bout number \times average duration, nor total distance traveled correlated with motor learning (see the new figure 3), whereas sleep amount did. Overall, this supports the conclusion that enhanced sleep, rather than differences in wheel use, was the key driver of improved motor learning.

See point 4 of Reviewer 1 for the applied changes.

10. In Results, section “Rocking during sleep promotes motor learning”, sentence: “A positive correlation was observed in both groups, but statistical significance was reached only in the SE group (SE: $r = 0.522$, $P = 0.045$; S: $r = 0.475$, $P = 0.12$, Figure 3E)”. This is small difference in both effect size and significance, so the authors may want to temper claims of group differences. Furthermore, showing significance in one group and not in another is a classic statistical error that should be fixed. Significance in one group but not another does not suggest that the groups are significantly different from each other. That should be tested formally, and one way, among a few, would be to test for a Group X Sleep interaction in predicting learning improvement.

We thank the reviewer for raising this point. To formally test whether the relationship between sleep and motor learning differed between groups, we fitted a linear model including a Group \times Sleep interaction. The interaction was not significant ($\beta = 30.68$, $SE = 152.15$, $t = 0.20$, $P = 0.84$ for learning estimated using maximum speed; $\beta = -30.5$, $SE = 131.58$, $t = -0.23$, $P = 0.82$ for learning estimated using average speed), indicating that the slopes did not differ between the SE and S groups. Accordingly, both groups exhibited a positive correlation, and we have revised the Results text to clarify that the statistical analysis does not support a group difference.

“...To formally assess whether these relationships differed between groups, we fitted a linear model including a Group \times Sleep interaction. The interaction was not significant (max: $\beta = 30.68$, $SE = 152.15$, $t = 0.20$, $P = 0.84$; avg: $\beta = -30.5$, $SE = 131.58$, $t = -0.23$, $P = 0.82$), indicating similar slopes between SE and S groups for both maximum and average speed. Together, these results suggest that longer sleep duration was similarly associated with greater motor learning in both conditions...”

11. In results, section “Improved motor learning is associated with increased transcription of synaptic genes”, what does “regulation of behavior” mean? That is vague and should be clarified.

We thank the reviewer for this comment. The term 'regulation of behavior' is a Gene Ontology (GO) term derived from the GO Biological Process ontology. We acknowledge that some GO terms may sound broad or vague, but they are widely used and accepted in the field to categorize gene functions.

12. “Thus, improved motor learning was associated with an increased density of glutamatergic synapses.” This claim should be tempered or directly tested by correlating density increases with learning. A group difference doesn't tell us that the density is related to motor learning. Similar claims throughout the Figures/Discussion should be similarly addressed.

To directly address this point, we performed a correlation between VGLUT1 puncta density and learning, assessed via both maximum and average running speed. While both learning metrics showed positive trends with VGLUT1 density, these correlations did not reach statistical significance (max: $r=0.44$, $p=0.2$; avg: $r=0.46$, $p=0.19$). Descriptively, SE mice tended to cluster toward the upper right (higher speed and more VGLUT1 puncta), while S mice were more represented in the lower left. In

light of this evidence, we have tempered our original claim and now refer to this relationship as a putative, non-significant association. We have modified the results section as follows:

Results

“Finally, to investigate whether the increased density of glutamatergic synapses in SE mice was directly associated with motor learning, we performed a correlation analysis between VGLUT1 puncta density in the motor cortex and learning performance, quantified as both maximum and average running speed on the complex wheel. While both metrics of motor learning showed positive trends with VGLUT1 density (maximum speed: $r = 0.44$, $P = 0.2$; average speed: $r = 0.46$, $P = 0.19$), these correlations did not reach statistical significance (Figure 6G-H).

Thus, rocking with motor learning was associated with an increased density of glutamatergic synapses.”

Figure 6...g-h. Correlation between learning (maximum (g) and average (h) speed) and VGLUT-1 puncta density in S and SE mice...

13. The authors correctly interpret their results and suggest that the synaptic changes in the SE group were caused by the increased motor learning. Could an alternative pathway be considered where sleep enhancement increases overall wheel-use (see above comment about average bout length vs total number of bouts), which then drives synaptic changes?

As noted above, we did observe an increase in average bout number in SE compared to S. However, overall measures of wheel use, including bout number \times average duration and total distance traveled, did not differ significantly between groups, despite SE mice showing slightly higher values. More importantly, none of these wheel-use metrics correlated with motor learning, whereas sleep amount did. This suggests that the synaptic changes observed in SE mice are more likely attributable to enhanced sleep facilitating motor learning, rather than increased wheel activity per se.

See also previous points (Reviewer 1 point 4; Reviewer 2 point 9).

14. In the Experimental Design section, and in the Results, it would be helpful to the reader if the authors could make explicit the hypothesis-testing logic for each of their three experiments? For example, experiment 3 is intended to dissociate sleep effects from wheel-running effects.

We thank the reviewer for this suggestion. We have revised the Experimental Design section and Results section to explicitly state the hypothesis tested in each experiment. These changes clarify the purpose and interpretation of each experiment.

15. In methods: “Sleep scoring was conducted offline by visual inspection of 4-second epochs using

SleepSign software. Behavioral state classification and further analysis were performed using custom MATLAB scripts.” Needs citation for validation of the staging.

We thank the reviewer for this suggestion. Sleep scoring using 4-second epochs has been widely validated for rodents (e.g., Cusinato et al., 2024, J Neurosci Methods. 2024 Aug;408:110155.) and is commonly used for manual classification of sleep/wake stages. We have now added this citation to the Methods section.

16. The authors should address Limitations more explicitly in the Discussion. For example, could there be a potential order effect due to having the baseline sleep (for the normalized sleep calculations) always come from the last 3 days of the habituation (prior to experimental phase)? Baseline data could have controlled for order by collecting data after the Rocking phase, and if this was not done, it should be discussed as a limitation as there is an unbalanced amount of time from the surgeries for each condition.

We thank the reviewer for highlighting this point. We acknowledge that using the last three days of habituation as baseline for normalized sleep calculations could introduce a potential order effect, as baseline measurements were always collected prior to the experimental phase. It is worth noting, however, that surgeries were performed only in a subset of mice used for pilot/polysomnography experiments, whose purpose was to validate the motion detection approach; no surgeries were performed in the other animals. Therefore, the time elapsed since surgery is not relevant for the majority of mice. Nonetheless, we recognize that the timing of baseline measurements relative to experimental manipulations could have introduced minor imbalances across conditions, and we have now explicitly acknowledged this limitation in the Discussion.

“Some aspects of the study should be considered when interpreting the results. First, baseline sleep measurements were always collected prior to the experimental phase, and their order was not randomized, which could have introduced minor order effects. Second, synaptic changes were assessed using immunofluorescence and confocal microscopy, providing reliable estimates of synapse density but not capturing ultrastructural or subtle changes in synaptic strength. Employing electron microscopy would allow higher-resolution analysis and a more detailed characterization of synaptic modifications. Third, although we controlled for differences in physical activity, other unmeasured factors, such as stress levels or wake quality, could have influenced the results. Lastly, only male mice were studied, limiting the generalizability of our findings across sex and age.”

17. Figure 1A should have some information about the duration in days/hours of the pilot study phases. The Results text also does not specify. That would be helpful instead of finding that detail in the Methods.

We thank the reviewer for this suggestion. The duration of the pilot study phases is already described in the Methods section: ‘In the polysomnographic experiment, EEG/EMG-implemented mice underwent a day of baseline followed by a day of rocking, with rocking active only during the light phase (8:00 AM-8:00 PM).’ We have reviewed the figure and text and believe that the information is clear; however, we have double-checked the labeling in Figure 1A to ensure it is fully visible and unambiguous.

18. Figure 2A could use a redesign to make the study phases clearer. Currently, it could be interpreted to mean 11 days of Rocking, then 11 days of wheel, as the yellow/grey shaded bars are not defined as the Light/Dark phases of the day. The time scale is also different for Baseline and the Rocking-on/Wheel, since Baseline is meant to show a 24hour period repeated 5 times, and the others are only showing 12-hour periods.

We have revised Figure 2A to improve clarity and avoid potential misinterpretation. The updated figure now clearly indicates the Light and Dark phases, defines the duration of each study phase, and aligns the time scales for Baseline and Rocking/Wheel periods, making the experimental design easier to follow.

19. From the video-monitoring, could the authors report the overall activity of the animals in each condition? Figure 2B looks like they had more activity during the Dark phase for the SE group, which alone could explain the motor learning and transcription effects. It could even potentially explain the motor learning and transcription effects, given the known connections between exercise, sleep, and plasticity.

We thank the reviewer for raising this interesting point. Motion detection during the dark period was less reliable than during the light phase because mice running on the wheel were partially obscured by the wheel itself, limiting accurate tracking. Consequently, when the wheels were present, motion detection could not reliably estimate sleep and wake. However, in the pilot PSG and third experiments, where no wheels were used, both EEG recordings (Figure 1d) and motion-based sleep/wake estimates (Figure below) confirmed that increased sleep during the light phase did not significantly alter time spent in wakefulness during the dark. Therefore, we do not believe that differences in dark-phase wake duration can account for the enhanced motor learning observed in the SE group relative to S. We cannot, however, exclude the possibility that the quality of wakefulness differed. We now report these findings as supplementary and we discuss them accordingly.

Supplementary Figure 1: Wake episode number and duration during the dark period in S (n = 5) and SE (n = 8) mice. Values are reported as normalized to baseline (last three days of the habituation period). Bars represent mean ± SD.

Results

“...with no effects on wake time during the dark phase (wake bouts: P=0.7; normalize wake duration: P=0.9, Supplementary Figure 1).”

Discussion

“We also considered the possibility that differences in wake duration during the dark phase might have influenced motor learning. However, both EEG recordings and motion-based sleep/wake estimates indicated that the amount of wakefulness during the dark period was similar in SE and S mice. Nevertheless, we cannot exclude the possibility that the “quality” of wakefulness differed between the two groups, which could, in principle, have affected learning.”

20. Figures 2C/3B, what are the error bars? Standard error of the mean? Please specify.

Fixed

21. In general, statistics reported in the text can be rounded to the first significant digit e.g. $p = 0.3574$ should just be 0.36.

We have revised the statistical values throughout the manuscript to ensure consistency, rounding them to the first significant digit as suggested. However, in cases where the P value was close to the significance threshold (e.g., $P = 0.045$) or was lower than 0.01 (e.g. = 0.002), we retained the third decimal to provide greater precision.

22. Figure 3B, is the Control data on Day 11 missing error bars?

Thanks for noting this. Now they are reported.

23. Figure 6 – I suggest adding perisomatic vs neuropil labels for each row, in addition to what is stated in the legend.

Corrected

24. Minor suggestion for Figure 7B – add labels for which one had rocking or not. I realize it’s meant to align horizontally with 7A, but that could be made more obvious.

Fixed

25. Overall, the statistical reporting would benefit from use of effect sizes, in addition to p-values.

To simplify the text and improve clarity, we have now included a comprehensive table (Supplementary Data file) reporting all statistical parameters, the tests used, effect sizes, confidence intervals, and exact p-values.

26. A possible confound, have the authors considered unintended heating by the rocking apparatus? If the apparatus is running 12 hours per day for 11 days, could this have any impact on the temperature of the chamber? If so, that is a confound that should be explicitly addressed, or better, repeat the experiment with controls for that. We know that temperature can affect sleep, so this detail may matter.

Thank you for raising this point. Empirically, the surface where the cages were placed did not feel warm to the touch after rocking. To verify this systematically, we monitored temperature using a digital thermometer placed inside the cages where the mice were housed. Ambient temperature was recorded over 24 hours during alternating periods of shaker ON and OFF across 30 days (15 days shaker ON + 15 days shaker OFF). Measurements were done at light ON at 8 AM and at light OFF at 8 PM. As expected, temperature showed small daily fluctuations, with slightly higher values at 8 PM

than at 8 AM (max range: 0.8°C), but always remained well within the standard range for mouse housing (20–24°C). Importantly, this variation did not differ between shaker ON and OFF conditions, and the condition effect was not statistically significant ($P=0.69$ from RM-2wayANOVA, $P=0.51$ from direct t-test). These results indicate that unintended heating from the apparatus did not occur—or, if present, did not reach the area where the mice were housed—and is therefore unlikely to have affected sleep.

Figure. *Left:* Ambient temperature measured at lights ON (8 AM) and lights OFF (8 PM) over 30 days (15 days with shaker ON and 15 days with shaker OFF). A slight increase in temperature was observed at 8 PM compared to 8 AM under both conditions. *Right:* Average ambient temperature did not differ between shaker ON and OFF periods.

Reviewer #3 (Remarks to the Author):

Major Claims

The manuscript investigates how vestibular stimulation (rocking) influences sleep and subsequent behavioral outcomes, with additional analyses on synaptic density markers. The central claims are that rocking modulates sleep-wake patterns and improves memory-related measures. While the idea of rocking as a modulator of sleep is intriguing and of potential translational interest, aspects of the claims lack novelty without strong mechanistic grounding. Several prior studies should be cited to contextualize rocking and vestibular contributions to sleep.

Convincing Evidence

The data provide preliminary evidence but do not convincingly support the full conclusions.

- The pilot PSG dataset is important but limited in scope; most conclusions are drawn from behavioral inferences that could be confounded.

We thank the reviewer for their insightful assessment and constructive comments.

The pilot PSG data were intended to confirm that rocking enhances sleep using a gold-standard approach. This effect has been demonstrated previously by other laboratories, including our own, using extensive EEG analyses. In the present study, our focus was limited to demonstrating that rocking is capable of extending sleep. Regarding potential behavioral confounds, we note that video monitoring has been thoroughly validated as a reliable proxy for sleep and wake estimation in mice, and previous studies have confirmed its accuracy. Therefore, we believe that the behavioral inferences drawn in our study are robust.

- Tapping during certain conditions acts as an uncontrolled arousal cue, complicating interpretation of rocking-specific effects.

We thank the reviewer for this comment. As described, tapping was chosen as the least stressful cue to keep the mice awake. Alternative interventions, such as introducing novel objects or gentle handling, would have been considerably more intrusive. Importantly, tapping was applied only during the first four hours of the dark phase in Experiment 2, was applied equally to both groups of mice, and in rare occasions, since mice are usually spontaneously active during the first part of the dark period. Given these considerations, we do not believe that this minimal intervention could have acted as a confounding factor affecting behavioral outcomes.

- Additional controls (e.g., sham vibration/noise, habituation analysis across days) and colocalization markers (VGLUT1 + PSD-95) would strengthen claims about synaptic plasticity.

We thank the reviewer for this constructive suggestion. We did not include noise or vibration as separate control groups because the rocking cage was placed in close proximity to the no-rocking cage, and any potential noise or vibration produced by the shaker (although imperceptible to humans) would have affected both groups of mice equally. Over 11 days of observation, we did not detect signs of habituation in this strain; however, we cannot exclude that longer periods of stimulation might induce some degree of habituation. For this reason, we did not further investigate habituation in the present study.

VGLUT1 and VGAT are well-established markers of excitatory and inhibitory synapses, respectively (Melone M et al. J Comp Neurol. 2005 Nov 28;492(4):495-509; Bragina L et al., Neuroscience. 2007 Jun 8;146(4):1829-40), and we reasoned that adding additional colocalization markers would not substantially increase the specificity of our findings. No single marker fully captures all synaptic subtypes; for example, PSD-95 is not expressed in all spines (Micheva KD, et al. Neuron. 2010 Nov 18;68(4):639-53), and its colocalization with VGLUT or VGAT would be limited since these are presynaptic markers, whereas PSD-95 is postsynaptic and restricted to glutamatergic synapses (see the image below for reference). Electron microscopy (EM) would provide the highest level of resolution; however, performing EM across all animals and conditions would have considerably prolonged the study and was therefore not feasible within the scope of this work. We now explicitly acknowledge this limitation in the Discussion.

Figure legend: VGLUT1 (red) and PSD-95 (green) co-staining in the cerebral cortex (Layer II-III).

Discussion

“...The observed increase in glutamatergic synaptic density in the SE group is likely the result of improved motor learning rather than the direct effect of sleep enhancement. Indeed, sleep enhancement in the absence of motor learning did not lead to changes in synapse number within the motor cortex. Although we cannot exclude the possibility of more subtle changes in synaptic strength, such effects may have gone undetected due to the limitations of confocal microscopy. Specifically, quantifying changes in the size of punctate staining through immunofluorescence lacks sufficient resolution and reliability for capturing nuanced modifications in synapse structure, which usually requires analysis at the electron microscopy⁴⁵⁻⁴⁷. Further investigation,

targeting the molecular and ultrastructural effects of rocking alone, will help determine whether this method of sleep enhancement also induces synaptic strength remodelling, as observed during baseline sleep^{1,48-50}.

... Some aspects of the study should be considered when interpreting the results. First, baseline sleep measurements were always collected prior to the experimental phase, and their order was not randomized, which could have introduced minor order effects. Second, synaptic changes were assessed using immunofluorescence and confocal microscopy, providing reliable estimates of synapse density but not capturing ultrastructural or subtle changes in synaptic strength. Employing electron microscopy would allow higher-resolution analysis and a more detailed characterization of synaptic modifications.”

Impact on the Field

If clarified and strengthened, this work could spark interest in non-pharmacological interventions to improve sleep and learning. However, as currently presented, the conclusions are not robust enough to significantly alter prevailing thinking in the field. The potential translational impact is promising but requires stronger mechanistic and methodological rigor.

We appreciate the reviewer’s perspective. We recognize that rocking in humans may be challenging to implement, although some studies have demonstrated its feasibility and beneficial effects (see for example: Vulturar DM, et al., J Pers Med. 2024 Feb 19;14(2):218., van Sluijs RM, et al., Front Neurosci. 2020 Jan 21;13:1446. Perrault AA et al., Curr Biol. 2019 Feb 4;29(3):402-411.e3.).

In our work, rocking was used as a means to enhance sleep through sensory stimulation, an approach that is non-intrusive and non-pharmacological. In the sleep field, sensory stimulation has been widely investigated to improve sleep quality; however, to our knowledge, no prior study has examined the molecular or synaptic changes associated with such interventions. While we acknowledge the limited immediate translational scope, we believe our study advances understanding of how sensory stimulation during sleep influences the brain, thereby providing a foundation for future mechanistic and translational research.

and

Statistical Analysis

Statistical approaches do not appear fully aligned with the design. Given repeated measures across the day and possible cage-level clustering, simple t-tests and ANOVAs are not sufficient. Mixed-effects models would be more appropriate. Effect sizes, confidence intervals, and exact p-values should be consistently reported. Without this, the strength of the evidence is difficult to assess.

We thank the reviewer for this important suggestion. When appropriate, we performed repeated-measures analyses and employed mixed-effects models appropriately accounting for within-subject dependence and unbalanced data. As all animals were single-housed, cage-level effects were not included in the models. To improve transparency and clarity, we have now included a comprehensive table reporting all statistical parameters, the tests used, effect sizes, confidence intervals, and exact p-values (see Supplementary Data file). This provides a more complete view of the analyses and allows readers to fully assess the strength of the evidence.

Reproducibility

The methodological description is detailed in some areas (e.g., rocking apparatus) but insufficient in others:

- Rocking parameters (1 Hz, ±20 mm displacement) are not justified or compared against alternatives.

We thank the reviewer for this comment. The choice of rocking parameters (1 Hz, ±20 mm displacement) was guided by a previous study (Kompotis K, et al., Curr Biol. 2019 Feb 4;29(3):392-401.e4.) that rigorously compared a range of frequencies and amplitudes. We have now cited this study in the text to justify our parameter selection and added a brief explanation of this rationale.

“Sleep enhancement was carried out by placing the mouse cage on a flat reciprocal shaker that allows horizontal movements (HS 260 Control, IKA, Switzerland). Rocking occurred at 1 Hz with a fixed peak displacement of ± 20 mm. These parameters were chosen based on prior evidence indicating that they maximize the sleep-enhancing effect while preserving normal sleep architecture.¹³”

- Housing conditions (group vs. individual) and cage placement on the platform could strongly affect results but are not thoroughly described.

We thank the reviewer for raising this point. In our study, all mice were single-housed, and we did not compare group versus individual housing. Details regarding housing conditions and cage placement on the platform are now clearly described in the Methods section (see Methods: “At 40 days of age (post-natal day (P)40), mice were selected for the experiment and housed individually in a cage composed of four separate chambers. Each chamber was a 25x25 cm square and hosted a single mouse. Chamber walls were transparent and equipped with small holes to facilitate the diffusion of smells and limit the effects of social isolation. In each chamber, mice were constantly video-monitored. Mice were allowed to habituate to this new environment for a week before starting the sleep manipulation.”).

Other Concerns

- Habituation is not considered; 11 days of exposure may lead to reduced efficacy after 7 days.

We appreciate the reviewer’s concern. However, we did not observe a decrease in rocking efficacy during the 11-day observation period. As noted above, we cannot exclude the possibility that longer stimulation periods might lead to habituation, but this was beyond the scope of the present study.

- Recovery sleep/homeostatic rebound may explain some effects rather than rocking itself.

We thank the reviewer for this observation. To clarify, no sleep deprivation was performed in this study; therefore, recovery sleep or homeostatic rebound is unlikely to account for our findings. Moreover, by analyzing sleep amounts during the dark period, when rocking was OFF, we did not detect changes in sleep, suggesting no compensatory adjustment in this phase.

Suggestions for Clarity and Organization

- Clarify the rationale for parameter choices (frequency, duration, displacement).

We provided a rationale for these parameters in the Methods section. They were chosen based on previous work that systematically tested different settings and identified those that maximized sleep enhancement while minimizing disruption (see Ref. n.13 Kompotis et al 2019). See also the previous point.

- Reorganize Results to emphasize consistent findings, grouping nonsignificant results succinctly.

We thank the reviewer for this suggestion. While we appreciate the idea of grouping non-significant results, we believe that presenting our two control experiments separately (one controlling for the role of rocking itself and the other for the role of sleep enhancement alone without learning) conveys a clearer message. Keeping them as separate paragraphs allows the distinct purpose and interpretation of each control to be fully appreciated.

- Expand the Discussion to explicitly address alternative explanations (arousal cues, circadian phase, homeostasis).

We thank the reviewer for this suggestion. We have expanded the Discussion to explicitly address alternative explanations for our findings. In particular, we discuss in greater detail the results of our control experiments and how they help to disentangle the specific effects of rocking from possible

confounding factors. These additions clarify the interpretation of our results and provide a more comprehensive consideration of alternative mechanisms.

Discussion

“An important consideration is whether the enhanced performance of SE mice could reflect differences in wheel engagement rather than the effects of sleep itself. Although SE mice initiated running bouts more frequently, the total time spent running and distance covered did not differ significantly between groups. Moreover, these measures did not correlate with learning outcomes, in contrast to the clear relationship observed between sleep amount and performance improvement. Thus, it is unlikely that the improved learning observed in SE mice was simply due to increased physical activity or greater task exposure; rather, the data support the hypothesis that sleep enhancement directly facilitated the acquisition of this complex motor skill.

We also considered the possibility that differences in wake duration during the dark phase might have influenced motor learning. However, both EEG recordings and motion-based sleep/wake estimates indicated that the amount of wakefulness during the dark period was similar in SE and S mice. Nevertheless, we cannot exclude the possibility that the “quality” of wakefulness differed between the two groups, which could, in principle, have affected learning.

Finally, to further exclude nonspecific effects of vestibular stimulation, we performed a control experiment in which rocking was applied exclusively during wakefulness. This manipulation did not improve motor performance, supporting the interpretation that enhanced motor learning in SE mice was mediated by improved sleep rather than by the direct effects of rocking or increased wake activity.”

Transparency

I prefer to remain anonymous for this review.

Reviewer #4 (Remarks to the Author):

I co-reviewed this manuscript with one of the reviewers who provided the listed reports. This is part of the Communications Biology initiative to facilitate training in peer review and to provide appropriate recognition for Early Career Researchers who co-review manuscripts.

Thanks. We fully endorse this initiative.

REPLY TO REVIEWER #3

Reviewer #3 (Remarks to the Author):

The authors have responded thoughtfully and extensively to my previous concerns, substantially strengthening the manuscript. Most major issues have been addressed. However, some responses rely on justification rather than additional evidence, and a few conceptual and methodological limitations remain that should be more clearly acknowledged.

I appreciate the authors' careful and comprehensive responses to my previous comments. The manuscript has been substantially strengthened, particularly through improved statistical reporting, clearer hypothesis testing, and more appropriately the synaptic and behavioral findings.

while the control experiments substantially limit nonspecific arousal or activity confounds, the absence of a dedicated sham-rocking condition modestly constrains inferences about vestibular specificity and should be acknowledged as such.

Overall, the authors have satisfactorily addressed the major concerns raised in the initial review. With minor clarifications regarding these remaining points, the manuscript represents a well-controlled and timely contribution to the literature on sensory sleep enhancement and experience-dependent plasticity.

We thank the reviewer for this clarification. We have now explicitly acknowledged the absence of a dedicated sham-rocking condition as a limitation in the Discussion, noting that while our control experiments substantially limit nonspecific confounds, sham stimulation would further strengthen inferences regarding vestibular specificity. To better contextualize this point, we also cite prior work showing that mice with defective otolith function fail to exhibit rocking-induced sleep enhancement, supporting a primary role for vestibular signaling in mediating these effects. As noted by the reviewer, however, this does not exclude the possibility that additional sensory inputs (e.g., noise or vibration) may contribute to the observed effects in our experimental setup.

Discussion

“...Second, although our control experiments substantially limit nonspecific arousal, activity-related, and wake-time confounds, the absence of a dedicated sham-rocking condition modestly limits strict attribution of vestibular specificity. While previous work showing that mice with defective otolith function fail to exhibit rocking-induced sleep enhancement supports a primary role for vestibular signaling¹³, additional sensory inputs (e.g., noise or vibration) cannot be fully excluded in the present setup...”